# Latent Diffusion Model for DNA Sequence Generation

**Zehui Li**[*]
Imperial College London
zehui.li22@imperial.ac.uk

**Yuhao Ni**[*]
Imperial College London
harry.ni21@imperial.ac.uk

**Tim August B. Huygelen**
University College London
thuygelen@gmail.com

**Akashaditya Das**
Imperial College London
akashaditya.das13@imperial.ac.uk

**Guoxuan Xia**
Imperial College London
g.xia21@imperial.ac.uk

**Guy-Bart Stan**[†]
Imperial College London
g.stan@imperial.ac.uk

**Yiren Zhao**[†]
Imperial College London
a.zhao@imperial.ac.uk

## Abstract

The harnessing of machine learning, especially deep generative models, has opened up promising avenues in the field of synthetic DNA sequence generation. Whilst Generative Adversarial Networks (GANs) have gained traction for this application, they often face issues such as limited sample diversity and mode collapse. On the other hand, Diffusion Models are a promising new class of generative models that are not burdened with these problems, enabling them to reach the state-of-the-art in domains such as image generation. In light of this, we propose a novel *latent diffusion* model, DiscDiff, tailored for discrete DNA sequence generation. By simply embedding discrete DNA sequences into a continuous latent space using an autoencoder, we are able to leverage the powerful generative abilities of continuous diffusion models for the generation of discrete data. Additionally, we introduce Fréchet Reconstruction Distance (FReD) as a new metric to measure the sample quality of DNA sequence generations. Our DiscDiff model demonstrates an ability to generate synthetic DNA sequences that align closely with real DNA in terms of Motif Distribution, Latent Embedding Distribution (FReD), and Chromatin Profiles. Additionally, we contribute a comprehensive cross-species dataset of 150K unique promoter-gene sequences from 15 species, enriching resources for future generative modelling in genomics. We have made our code and data publicly available at https://github.com/Zehui127/Latent-DNA-Diffusion.

## 1 Introduction

Designing synthetic DNA sequences for genetic modifications is traditionally guided by organism-specific workflows derived from extensive laboratory experiments. As the amount of data generates from these workflows expand, deep generative models are well-positioned to enable a new frontier in synthetic DNA sequence generation over a wide range of potential applications [12, 37]. Generative adversarial networks (GANs) [11] are a popular choice for the generation of synthetic DNA sequences, as demonstrated by the studies of [18, 34, 37]. GANs can effectively generate sequences, however, it has been shown that the generated samples lack diversity [8] and suffer from mode collapse at training time [22].

---

[*]Equal contribution.

[†]Correspondence should be addressed to: {zehui.li22,harry.ni21,g.stan,a.zhao}@imperial.ac.uk.

NeurIPS 2023 AI for Science Workshop.

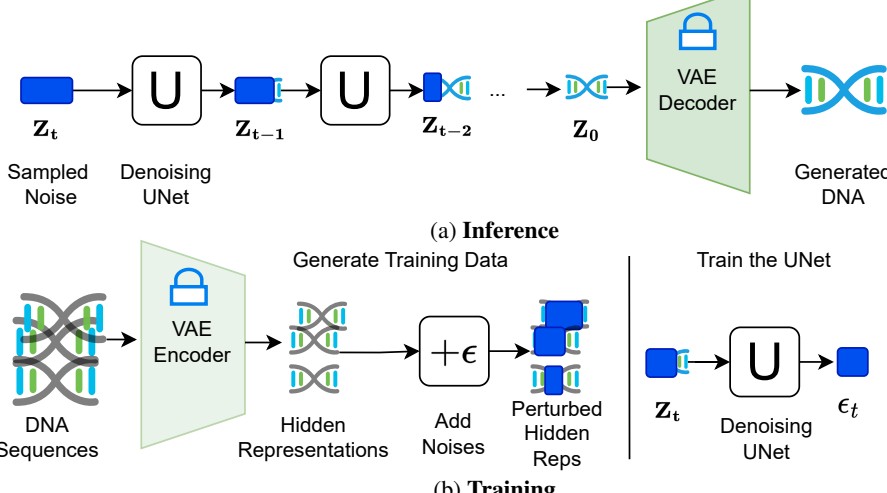

(a) **Inference**

(b) **Training**

Figure 1: **Denoising View of Latent Diffusion Models for DNA sequences.** (a) A zoom-in view of the inference process. (b) A zoom-in view of the training process. The blue square represents the noises attached to the latent representation of the DNA sequences.

Given the success of diffusion models in image generation [26], protein synthesis [36], and circuit design [35]. The application of diffusion models for DNA sequence generation may generate sequences of higher quality. In this case, quality is a combination of sequence diversity and the ability to capture the underlying distribution/motifs. In this paper, we propose a latent diffusion model for discrete data generation and apply it to DNA sequence generation. Our contributions are as follows:

**A generalisable framework for discrete data generation:**    We propose a latent diffusion model made from a transformation function and a denoising model. It separates the learning of latent distribution and the denoising model. A lightweight Variational Auto-encoder architecture is used for the transformation function for 1-dimensional discrete sequences.

**A generative model for DNA Sequence Generation:**    We apply the proposed latent diffusion model to DNA sequence generation, obtaining promoter-gene samples with the properties of realistic DNA sequences across different species.

**An evaluation metric for generated DNA sequences:**    We introduce Fréchet Reconstruction Distance (FReD) as a numerical metric to evaluate the quality of the generated DNA sequences. FReD is consistent with motif distribution, the standard qualitative approach to evaluate the generated DNA sequences.

**A cross-species dataset for DNA sequence generation:**    We curate and create a dataset containing 150K unique promoter-gene sequences across 15 species. This dataset paves the way for the construction of a generative model for promoter sequences

## 2   Background

**Diffusion Models**    Intuitively, Diffusion Models (DMs) incrementally construct realistic examples from sampled noise. At each time step $t$, a denoising function is used to predict a more realistic example $x_{t-1}$ from $x_t$. An alternative perspective by Song et al. [31] uses stochastic differential equations (SDE) to describe the diffusion models. This unifies previous scored-based models [30] and DDPM [15], providing a powerful formulation for analyzing DMs.

In [31], the process of adding noises is modelled with an SDE with a Wiener process:

$$\mathrm{d}x = f(x, t)\,\mathrm{d}t + g(t)\,\mathrm{d}w \tag{1}$$

- The function $f$ is called the drift coefficient. This indicates how the process $x$ tends to evolve over time, ignoring any random fluctuations. Multiplying by $\mathrm{d}t$ scales the drift by the time increment.

- $g(t)$ is called the diffusion coefficient. It measures the volatility or variability of the process. It only depends on time $t$.

- $\mathrm{d}w$ represents the increment of the standard Wiener process (or Brownian motion) $w$ over an infinitesimally small time interval.

The reverse process, which generates the realistic samples, can be represented by another SDE:

$$\mathrm{d}x = [f(x,t) - g(t)^2 \nabla_x \log p_t(x)]\,\mathrm{d}t + g(t)\,\mathrm{d}\overline{w} \tag{2}$$

Where $\mathrm{d}\overline{w}$ represents the increment of the reverse Wiener process. $\nabla_x \log p_t(x)$ is the gradient of the log Probability Density Function (PDF) of $x$ at time $t$. At the training time, a score function $\mathbf{S}_\theta(x(t), t)$ is trained to approximate $\nabla_x \log p_t(x)$; and then realistic samples can be generated by walking through Equation (2).

**Diffusion Models for Discrete Data**    There are two reasons why standard DMs cannot deal with discrete data ($x$): 1) the diffusion process with the Wiener process defined by Equation (1) is described in terms of continuous variables, it is undefined when $x$ is discrete. 2) the solution led by the reverse process requires a differentiable PDF $\log p_t(x)$ to exist, but discrete variable $x$ does not have a PDF.

To account for these challenges, two possible solutions can be used to develop DMs for discrete data. They are 1) define a diffusion-like process to model discrete data [1, 3, 32]. 2) map the discrete input into a continuous latent space [9, 13, 19]. For more details, see Appendix A.

Differing from the prior work on latent DMs for discrete data [9, 33], which is trained in an end-to-end manner, we aim to separate learning of the mapping from discrete to the latent space and score function. This facilitates the learning task by not learning the latent distribution and score-based denoising model at the same time. It also removes the need to adjust the weight of reconstruction and generation in the end-to-end training [26].

## 3   DiscDiff: A Latent Diffusion Model for Discrete Data

We introduce DiscDiff, a flexible latent diffusion model designed for discrete data generation. This model is structured into two main components: a transformation function and a denoising model. The transformation function is implemented with a lightweight Variational-Auto-Encoder (VAE), with an encoder $z = \mathbf{E}(x)$, which translates discrete input $x$ to a continuous latent variable $z$, and a decoder $\tilde{x} = \mathbf{D}(z)$, which reverts $z$ back to its discrete form $\tilde{x}$. The denoising model $\mathbf{S}_\theta(z(t), t)$ is employed to learn the score function $\nabla_z \log p_t(z)$ for the latent variable.

In the training process, the learning phases of the transformation function and the denoising model are separated. The first phase focuses on learning the transformation function $\tilde{x} = \mathbf{D}(\mathbf{E}(x))$, with a primary goal of minimising the reconstruction loss for discrete variable $x$. The second phase concentrates on training the score function estimator, aiming to minimise the difference between the estimator and $\nabla_z \log p_t(z)$.

### 3.1   Transformation Function

**Architecture**    In order to map discrete DNA data to continuous space, we employed a lightweight Variational Autoencoder (VAE) that employs a transformation function composed of Convolutional Neural Networks (CNNs). The structure of the VAE is depicted in Figure 2, illustrating how the latent variable is modelled with a multivariate Gaussian distribution. The discrete input passes through the encoder, through multiple layers designed to increase the channel dimension while reducing the length dimension. Next, a Conv2D block is used to extract essential features from the channel-length surface, mapping the input to a higher-dimensional space. In symmetry to the Encoder, the Decoder is constructed with transConv and upsampling operations, effectively inverting the encoding process.

The proposed VAE stands out for its lightweight nature. This is attributed to the CNN-based architecture which enables it to efficiently extract information from 1D discrete data and map it to

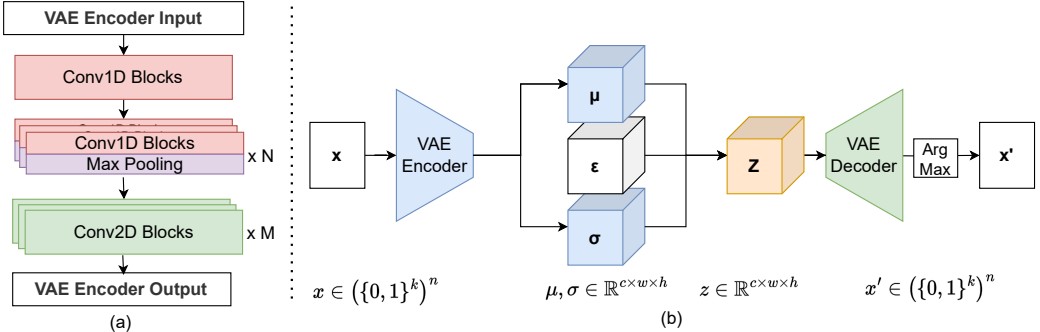

Figure 2: VAE as a transformation function. (a) shows the components of the encoder, consisting of conv1d and conv2d. It lifts the channel dimension and reduces the length dimension of the input. (b) shows the overall view, the latent variable is modelled with Diagonal Gaussian Distribution. The Decoder is symmetric to the Encoder architecture, consisting of transConv and upsampling. See Appendix D for the values of $c, w, h, \ldots$..

a high-dimensional continuous space. Additional details about the VAE architecture, including the incorporation of a multi-kernel design [7] in the Conv1D block for feature extraction at various scales, are provided in Appendix D.

**Cross Entropy Loss** When training the VAE, we propose to use Cross Entropy (CE) as reconstruction loss. The loss function is given by:

$$\mathbf{L}_{\theta,\phi} = \mathbb{E}_{z\sim q_\phi(z|x)} \underbrace{\left[ -\sum_{t=1}^{k}\sum_{l=1}^{n} p(x_t, l) \log p_\theta(x_t|z) \right]}_{\text{Reconstruction Loss}} + \mathbb{E}_{x\sim p(x)} \left[ \underbrace{\text{KL}(q_\phi(z|x) \,||\, \mathcal{N}(z; \mu, \Sigma))}_{\text{KL Divergence}} \right] \quad (3)$$

where $x \in \mathbb{R}^{k\times n}$ is onehot-encode data; $p_\theta(x|z)$ is the probabilistic decoder output from $\mathbf{D}_\theta$; $q_\phi(z|x)$ is the probabilistic output from encoder $\mathbf{E}_\phi$ that represents the approximate posterior of the latent variable $z$ given the input $x$; $\mathcal{N}(z; \mu, \Sigma)$ is the prior on $z$, here we use diagonal Gaussian Distribution.

## 3.2 Diffusion in the Latent Space

Once the transformation function with $\mathbf{D}_\theta$ and $\mathbf{E}_\phi$ are trained in the first stage, the diffusion model is applied to the latent representation $z = \mathbf{E}_\phi(x)$. With the goal of minimizing the difference between the estimator and the score function in the latent space.

$$min(\hat{\mathbf{S}}_\theta(z(t), t) - \nabla_z \log p_t(z))^2 \quad (4)$$

It is trained by sampling $t$ from uniform distribution $U[1, T]$ and $z_t$ from $p_t(z_t|z_0)$ with a known transition kernel defined by Equation (1). The training process can also be interpreted as a training a noise predictor [26]:

$$\mathbb{E}_{z,t\sim U[1,T],\varepsilon\sim\mathcal{N}(0,1)} \left[ \|\varepsilon - \varepsilon_\theta(z(t), t)\|_2^2 \right] \quad (5)$$

We adopted the UNet backbone from LDM [27] to implement $\varepsilon_\theta(z(t), t)$. The UNet can capture coarse-grained features and generate samples with high qualities in various domains [16, 20]. Figure 1 illustrates the denoising view of training and inference processes for the diffusion model. The details about the UNet configuration can be found in Appendix E.

## 3.3 Justification for the Two-stage Training

The separation of VAE and the denoising model training can be justified by the loss function of LSGM [33]. LSGM jointly trains a VAE and the denoising model with a training loss consisting of a reconstruction term and KL divergence between encoder distribution $p(z_0|x)$ and the prior distribution of the latent variable $p(z_0)$. The latter term can be expressed in terms of the score function $\mathbf{S}_\theta(x(t), t)$.

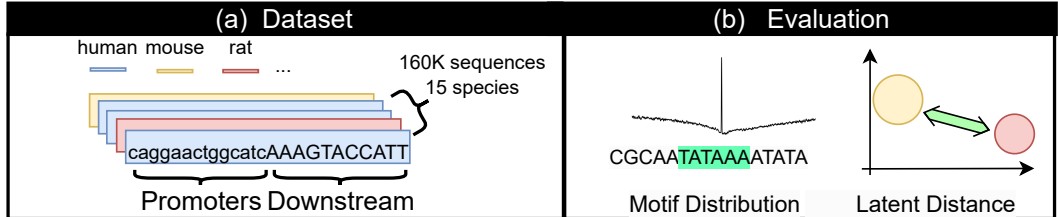

Figure 3: **Dataset and evaluation methods for DNA sequence generation.**(a) shows the format of the curated dataset from EPDnew (b) Motif Distribution compares the distribution of subsequences in DNA and Latent Distance metrics compare the distribution of latent embedding of DNA.

The optimisation of this loss is challenging, requiring different weighting mechanisms for the training objective and variance reduction techniques for stability. The separation of training is equivalent to a special training schedule with the KL divergence term being fixed in the first stage, and then training the denoising model in the second stage. The performance gain of separating the training process has been shown in the image generation domain [25, 26]. In the section below, we provide the empirical evaluation of this method for DNA sequence generation.

## 4  DNA Sequence Generation with DiscDiff

We explored the unconditional generation of both regulatory elements and protein-encoding genes. Our problem setting is different from the previous study DDSM [1], where the transcription profile is provided as an input at both training and inference time. In this study, we focus on a scenario where no transcription profile is given. The generative model needs to discover the properties of regulatory elements and genes purely from DNA sequences. Identifying and generating regulatory elements is essential for successful genetic modifications [4]. To achieve this, we constructed a dataset, trained the diffusion model for DNA generation, and provided an evaluation metric called Fréchet Reconstruction Distance (FReD) to assess the quality of generated samples. For an in-depth look into the training specifics, including hyperparameters and training equipment, see Appendix B.

### 4.1  Data Construction and Representation

**Dataset**    We created a cross-species dataset for DNA generation from the refined Eukaryotic Promoter Database (EPDnew) [10, 23], housing organism-specific promoters annotated with downstream genes, transcription starting sites, and expression levels. From EPDnew, we processed 160K unique sequences across various species, each mapping to different cell types and expression levels, detailed in Table 2 of Appendix C. This study emphasizes unconditional generation using a subset of this data. We employed a 2048-base context window centred on a gene's transcription starting site. Each sequence split evenly into an upstream promoter and a downstream DNA segment, spans 2048 bases in total.

**Representation of DNA Sequences**    DNA sequences are composed of the nucleotide bases $A$, $T$, $G$, and $C$. For a DNA sequence of length 2048, a one-hot encoding function, denoted as $f$, is employed:

$$f : \{A, T, G, C\}^{2048} \to \mathbb{R}^{4 \times 2048}$$

It is worth noting that DNA sequences sometimes contain the $N$ token, indicating an unknown or ambiguous nucleotide. In the encoding scheme, such a token is represented as $\{0.25, 0.25, 0.25, 0.25\}$. This representation allows generative models to directly model the aleatoric uncertainty arising from the data collection process when using a cross-entropy reconstruction loss for VAE training.

### 4.2  Evaluation Metrics

**Motif Distribution**    A motif in the context of molecular biology refers to a short sequence of DNA or RNA that has a specific structure or function. Motif distribution is a commonly used metric to

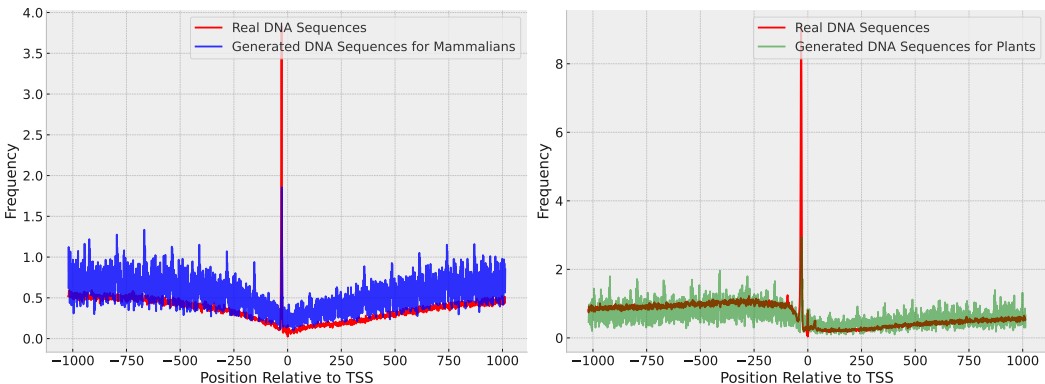

Figure 4: **TATA-Box Distribution in Real and Generated DNA Sequences**. The left plot represents mammalian sequences, while the right showcases plant sequences. Both generated samples closely mirror the distribution of their respective real DNA sequences.

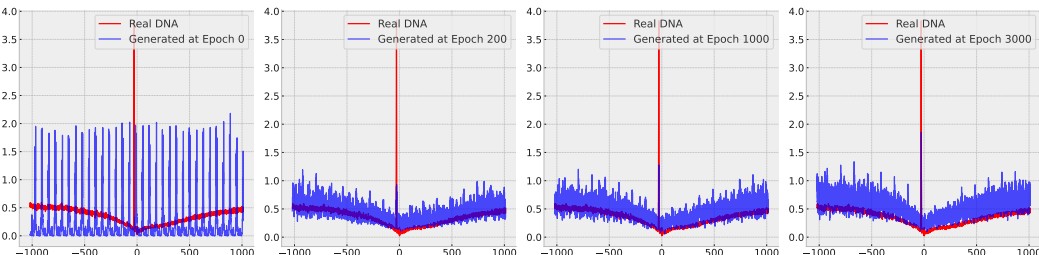

Figure 5: **TATA-Box Distribution Across Epochs Relative to Real DNA**. Subplots from left to right showcase the motif distribution by DiscDiff at epochs 0, 200, 1000, and 3000. With increasing epochs, the peak distribution around TSS converges towards that of the real DNA, but the background distribution starts to diverge after 200 epochs.

evaluate the biological relevance of generated DNA sequences, which is frequently employed in prior studies [1, 18, 34, 37]. DNA motifs with biological functions can shed light on the functionality of generated sequences. By comparing the distribution of motifs between real and generated sequences, we can measure the ability of the generative model to replicate the underlying patterning of natural DNA sequences. A high-quality generative model should exhibit a motif distribution closely mirroring that of genuine DNA sequences, ensuring that generated sequences retain crucial biological signatures.

**Fréchet Reconstruction Distance**    We introduce the Fréchet Reconstruction Distance (FReD) to assess the quality of generated DNA samples quantitatively. Unlike the Fréchet Inception Distance used for image generation evaluation [14], FReD leverages the encoder of a pre-trained Auto-Encoder (AE) to transform DNA sequences into embeddings. This encoding process measures the disparity between generated sample distributions and real-world sequences. For this purpose, we train an AE, following the architecture outlined in Section 3, on a reference genome distinct from the training data used for generation. The embeddings are subsequently derived from this encoder.

**Evaluating Generated Sequences with the Sei Framework**    Sei [5] is a deep-learning framework dedicated to predicting the chromatin profile of the human genome. We used Sei to evaluate the generated sequences in two ways. First, we employ Sei to encode the sequences and subsequently calculate the distribution distance. As highlighted in Section 5, there's a consistency between the results of FReD and Sei. Second, using Sei, we obtained predictions for 21,907 distinct chromatin profiles $Sei(x) \in \mathbb{R}^{50000 \times 21907}$ for 50,000 generated sequences. For each profile, we calculated the number of "hits" among our sequences. A "hit" is defined as a sequence for which the predicted likelihood of aligning with a profile exceeds 0.9.

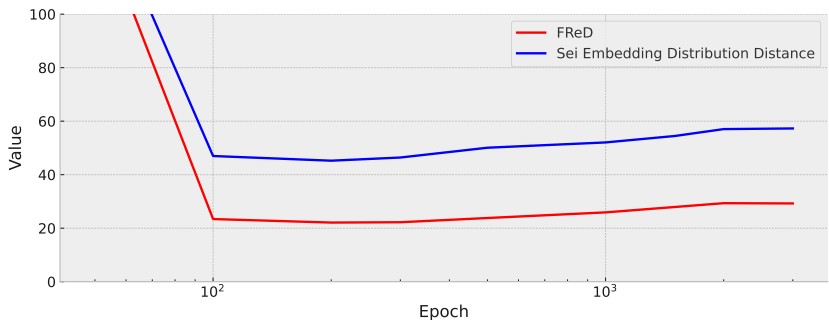

Figure 6: **Change of FReD and Sei Embedding Distribution Distance across epochs.** Both metrics show a rapid decrease until epoch 200, then gradually rise until epoch 3000. This behaviour indicates the complexity of capturing DNA sequence quality with a single value. Despite improvements in modelling the TSS peak, divergences in background motifs appear over extended training.

| Model | FReD↓ | Sei Embedding Distribution Distance↓ |
|---|---|---|
| Random | 251.7 | 250.7 |
| Sample from Training Set | 3.38 | 0.10 |
| VAE | 48.66 | 84.77 |
| DiscDiff 200 | **22.10** | **45.23** |
| DiscDiff 3000 | 29.25 | 57.28 |

Table 1: Comparison of FReD and Sei Embedding Distribution Distance across different models.

## 5   Results

**Motif Distribution**    To assess the quality of the generated samples, we generated 50,000 DNA sequences for both mammalian and plant species using DiscDiff. Their motif distributions are illustrated in Figure 4. The plots reveal consistency between the TATA-box [17] distributions of real DNA sequences and our generated promoters. Additionally, Figure 5 demonstrates the evolution of the motif distribution throughout training. Noticeably, while the peak distribution around the Transcription Starting Site (TSS) is converging to the real DNA sequences, the background distribution seems to be diverging after 200 training epochs. This trend is also captured by FReD and Sei Distance.

**Latent Distribution Distance**    Figure 6 presents the change of FReD and Sei Embedding Distribution Distance values relative to the training set across epochs. Notably, these metrics exhibit strong correlation to the training set: a sharp decline is observed from epoch 0 to 200, followed by a gradual increase up to epoch 3000. This trend highlights the intricacy of measuring the quality of generated DNA sequences using a singular numerical metric. We attribute the rise in these metrics (epoch 200 to 3000) to divergences in the background motif distribution. Even as the modelling of the TSS peak improves with prolonged training. Embedding-based approaches tend to prioritize the holistic representation of DNA sequences over specific details. However, latent distribution distances remain crucial as they help distinguish genuine DNA from random or subpar sequences. As per Table 1, when comparing VAE and DiscDiff, VAE-generated examples fare less favourably in motif distribution (Appendix F).

**Chromatin Profiles of Generated Sequences**    Figure 7 presents the chromatin profiles of the 50,000 generated and real DNA sequences. The y-axis indicates the count of sequences corresponding to each profile. Among these, we highlight the top 10 profiles with the highest counts, omitting cell line names for clarity. There's a striking resemblance between the generated sequences (Figure 7a) and the training data (Figure 7b) in terms of distribution and top-ranking profiles. Notably, profiles such as H3K4me3, H3K27me3 and H3K9me3 are predominant. H3Kxxme3 markers are closely linked to promoter activity as they remodel chromatin to be more accessible to transcription factors (essential proteins for promoter regulation). This observation aligns with our expectations since our training set is derived from the promoter database.

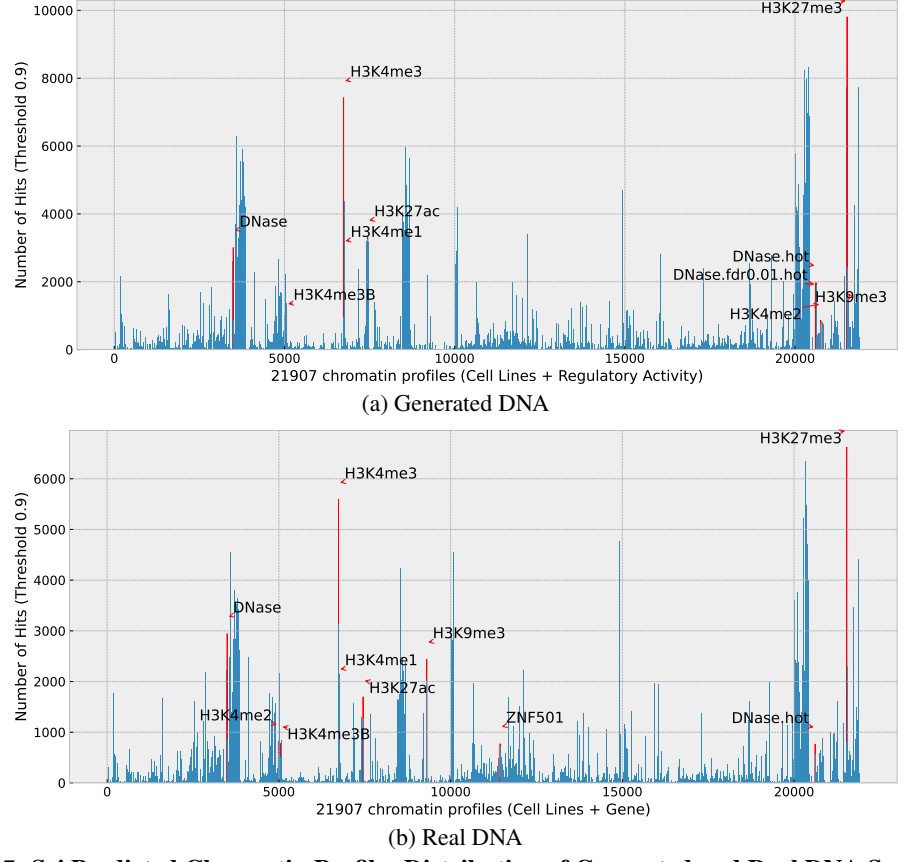

Figure 7: **Sei Predicted Chromatin Profiles Distribution of Generated and Real DNA Sequences.** Chromatin Profiles of (a) Generated DNA are aligned well with (b) Real DNA. The y-axis indicates the count of sequences for each profile, with the top 10 profiles highlighted. Predominant profiles like H3K4me3, H3K27me3, and H3K9me3, are linked to promoter activity.

## 6 Discussion

We presented DiscDiff, a latent diffusion model for DNA sequence generation. By connecting discrete DNA sequences to continuous spaces with an autoencoder, DiscDiff taps into diffusion model capabilities. Our Fréchet Reconstruction Distance (FReD) offers improved genomics evaluation, and our model's alignment with real DNA benchmarks showcases its efficacy.

Looking ahead, our approach raises important questions: How might we design general transformation functions optimally for this framework? What properties make a function particularly appealing for the diffusion model, ensuring the production of superior-quality samples? It has been shown in image generation, that the choice of auto-encoder will influence the quality of generated samples [26]. On the practical side, exploring the conditional generation of DNA sequences influenced by factors such as cell type, expression level, and environment holds potential. We envision a scenario where one could generate a functional gene or regulatory element based solely on specified criteria. In addition, the scarcity of experimental data for certain types of DNA sequences advocates the strategy of fine-tuning pre-trained diffusion models with existing methods [2, 28]. Such advancements could redefine synthetic biology, and we are optimistic about delving deeper into these avenues.

## Acknowledgements

This work was performed using the Sulis Tier 2 HPC platform hosted by the Scientific Computing Research Technology Platform at the University of Warwick, and the JADE Tier 2 HPC facility. Sulis is funded by EPSRC Grant EP/T022108/1 and the HPC Midlands+ consortium. JADE is funded by EPSRC Grant EP/T022205/1. Yuhao Ni acknowledges the undergraduate research opportunity

program award provided by his department in support of this research project and would like to express his sincere gratitude to Zehui Li, and Dr. Yiren Zhao for their guidance during the course of this research, as well as his family for their unwavering support. Zehui Li acknowledges the funding from the UKRI 21EBTA: EB-AI Consortium for Bioengineered Cells & Systems (AI-4-EB) award, Grant BB/W013770/1. Zehui Li acknowledged the support from his family and Ms. Yiqiu Sun.

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

## A  Extended Related Work of Discrete Diffusion Models

Two possible solutions has been used to develop DMs for discrete data. The first type of method defines diffusion-like processes to model the discrete data [3, 32]. The Dirichlet Diffusion Score Model proposed by [1] uses the Dirichlet distribution as the stationary distribution in the diffusion process and generates regulatory elements in DNA sequences.

The second approach is to map the discrete input into a continuous latent space. This approach has been used by several works for language modelling [9, 13, 19], where text is mapped to a continuous space through word2vec [24] or embedding obtained by neural networks. BitDiffusion [6] also

falls into this category, where discrete data is transformed into analogue bits and then real numbers. There are several advantages of mapping discrete data into continuous space. For example, effective sampling methods [21, 29] to accelerate the generation of DMs are still applicable. Moreover, compression is allowed when mapping the discrete data into the latent space, reducing the computational overhead for both training and inference of DMs.

## B    Training Details of Latent Diffusion Model

The model was trained utilizing the NVIDIA RTX A6000 graphics card. During the first stage, a Variational Autoencoder (VAE) was trained on a dataset comprising 160,000 sequences spanning 15 different species. This training was executed over 24 GPU hours with the primary goal of accurately reconstructing the input sequences. The chosen learning rate was 0.0001, paired with the Adam Optimizer. A batch size of $128$ was set, and the KL divergence weight was determined at $10^{-4}$. The resultant hidden representation, $z$, belongs to $\mathbb{R}^{16 \times 16 \times 16}$, which is precisely half the dimensionality of the original input $x \in \mathbb{R}^{4 \times 2048}$.

In the second training phase, a denoising UNet was trained on three GPUs across $3,000$ epochs. Our empirical observations highlighted that the UNet's performance improved with increased size. As such, given the GPU RAM capacity of $49$ GB, we optimized the model by maximizing the number of residual blocks and self-attention layers. The training also incorporated a cosine learning scheduler, introducing a warm-up period for the initial $500$ epochs. The learning rate was set at $0.00005$, and a batch size of $256$ was chosen for the training. DDPM [15] is used for sampling.

For a focused evaluation of the model's performance, particularly in mammalian DNA sequence generation, we opted for a subset of the entire dataset. This subset includes sequences from *H. Sapiens* (human), *Rattus Norvegicus* (rat), *Macaca mulatta*, and *Mus musculus* (mouse), which collectively represent 50% of the total dataset. Training on this subset not only streamlines the evaluation process but also allows for a more precise assessment of the generative algorithm's efficacy and accuracy in producing mammalian sequences

## C    Dataset Details

Table 2: The statistics of EPDnew dataset.

|  | # Unique Values | Examples |
|---|---|---|
| genes | 130,014 | CAACAAGGCACAGCAC... |
| promoters | 159,125 | GGCCCGGTTTTTTAAT... |
| cell types | 2713 | [Esophageal Epithelial Cells, Hippocampus, ...] |
| gene function description | 130,014 | [AHSP:alpha hemoglobin stabilizing protein, ...] |
| species | 15 | [Homo sapiens (Human).,Zea mays (corn)., ...] |
| expression levels | 29,349,475 | [250,264,338, ...] |
| transcription starting site | 29,349,475 | [10,0,-1, ...] |

**Source Description**    The European Promoter Database (EPD) is a vital tool for biologists, with a primary focus on gathering and categorising promoters of eukaryotes. The EPD comprises promoter data sourced from numerous published articles. The EPDnew database, or High-Throughput EPD (HT-EPD), provides an even more comprehensive and rigorous dataset. The EPDnew has been enhanced by combining conventional EPD promoters with in-house analyses of promoter-specific high-throughput data, albeit for only selected organisms. This strategy enhances the precision and commendable coverage of EPDnew. It's worth noting that the evidence used in EPDnew primarily arises from Transcription Start Site (TSS) mapping derived from high-throughput experiments, notably CAGE and Oligocapping.

**Data Selection**    Data was obtained from the EPDnew database without any exclusions, allowing for the incorporation of all available samples and associated promoters. Sequences between -1024 and +1023 base pairs from the transcription start site were chosen, ensuring a total length of 2048 base pairs - a convenient choice due to its alignment to powers of 2. This range covers crucial promoter

elements like GC and TATA boxes, and also allows the algorithm to identify any possible hidden patterns and interconnections preceding and succeeding the transcription start site.

**Time Frame**    The temporal aspect of our data is not limited to a specific time frame or year. The data was collected entirely as of August 2023.

**Data collection and preprocessing**    Data collection started by manually downloading data files through FTP, followed by accessing fasta data via the website's graphical interface. Carefully extracted metadata from Python scripts was then integrated with promoter sequences to form a comprehensive table.

Given the species-specific fragmentation of the acquired data, a iterative method aided in bringing together diverse FASTA files into one consistent table, with important attributes such as 'kingdom' and 'species' integrated into the DataFrame at extraction. A crucial pre-processing step involved categorising DNA sequences as 'upstream' or 'downstream' based on their relative positioning. Normalisation and transformation techniques were implemented where necessary to ensure data uniformity and integrity were maintained.

Simultaneously, an additional dataset containing gene expression data underwent similar aggregation processes. Promoter expression sample data was synchronised with the main DataFrame. To optimize data storage and retrieval, the processed data was archived as a Pickle file.

Data integration and duplicate removal processes were also undertaken. Subsequently, the integrity and consistency of the data have been validated through multiple checks, including the verification of value counts for specific columns and a comparison of the data against EPDnew statistics.

**Data Variables**

- Promoter Sequences: -1024 to 1023 bp from tss
- PID: A unique identifier for each promoter in the EPDnew database.
- Species: one of 15 species in our dataset
- Average Expression Value: mean of expression values of every sample pertaining to a specified EPDnew promoter
- Sample Name: Encompasses details like cell line and environmental conditions.
- Gene Description: description of gene function of the gene the promoter is regulating

**Data Size and Structure**    As shown in Table 2, the data set we compiled from EPDnew comprises of 159,125 distinct promoter sequences derived from 15 eukaryotic species, which are connected to 130,014 individual genes. Most of these genes have gene function definitions associated with them in the gathered data. The promoters have experimental expression levels and co-determined transcription start sites associated with them. These are frequently recorded in various cell types per promoter and might hold many experimental values for a single cell type. Our methodically structured dataset is in tabular form, facilitating smooth and comprehensive accessibility and manipulation for in-depth analysis.

# D    VAE Architecture

The convolutional VAE follows the encoder-decoder architecture. The encoder learns how to encode the discrete DNA sequence data into a continuous latent representation. Simultaneously, the decoder learns how to decode the latent representations into DNA sequence data in continuous space before it is quantised with ArgMax. A detailed illustration of the encoder architecture is in figure 8.

The decoder employs an architecture in symmetry with the encoder. In the decoder the same network layers are performed in reverse order and inverse operations are used in place of forward operations. Such inverse operation pairs include transposed convolution and convolution operation, upsampling and maxpooling operation.

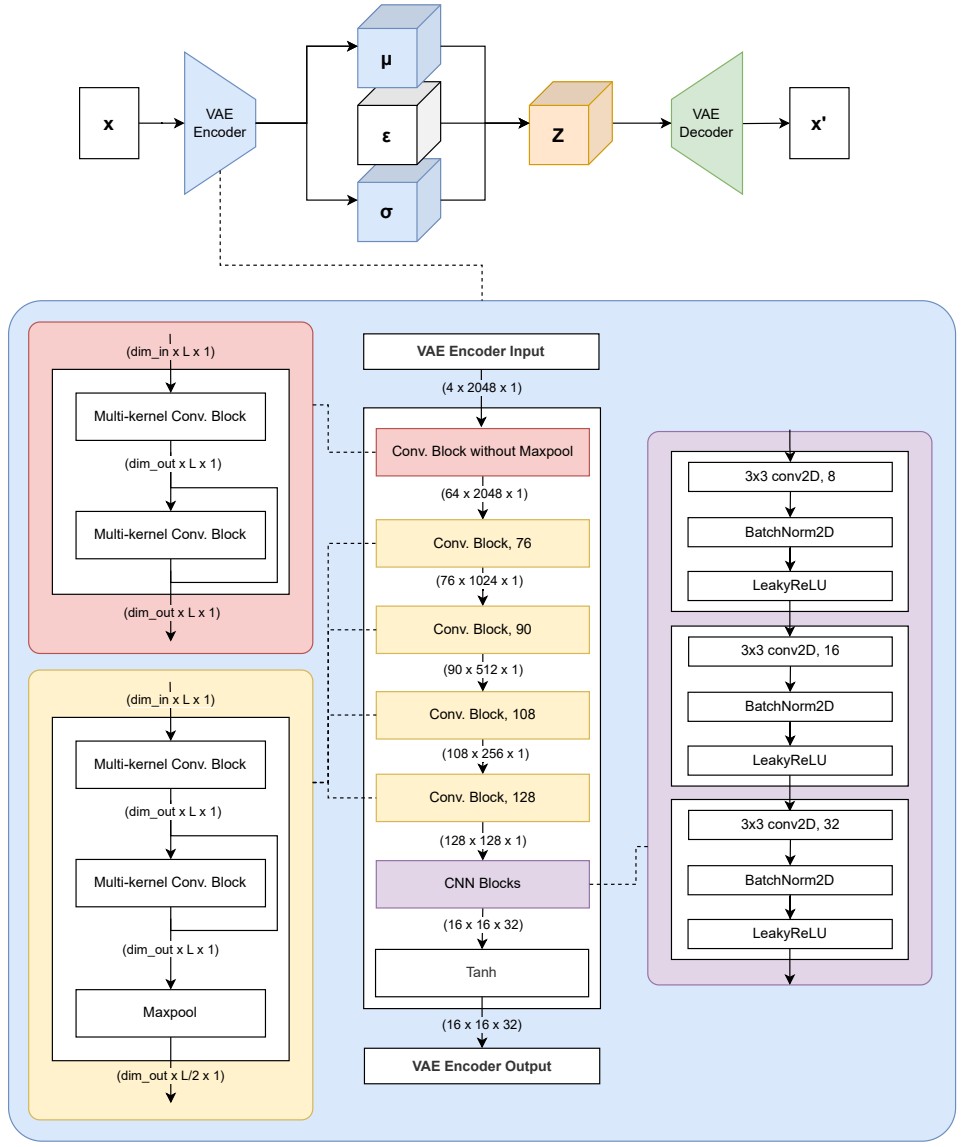

Figure 8: **VAE Encoder Architecture**
Two sections can be distinguished from the architecture: 1) the discrete data encoder (in red and yellow) and 2) the CNN Blocks (in purple)

The first section of the encoder encodes the one-hot encoded input into a 2-dimensional image-like representation. From the initial 4 channels due to one-hot encoding, the channel dimension is first increased to 64, then it increases further with steps of exponentially increasing number of kernels up to 128. This is implemented using Multi-kernel Convolution Blocks with perturbable kernel parameters combined with residual connections. By including a max-pooling layer in each layer, its length shrinks by powers of 2 to 128. This effectively expands our 4x2048x1 data into 128x128x1, an image-like surface when viewed along the channel and length. This stage of the encoder aims to capture high-level positional features along the length dimension via pooling, for each channel dimension of the DNA sequence by transforming from 1-dimensional features into a 2-dimensional surface.

The second section of the encoder uses a CNN architecture for learning latent representations. There are three layers of the CNN block, each performing 3x3 kernel on the channel-length surface followed by BatchNorm2D and LeakyReLU activation. The CNN block in the encoder learns to produce

different feature maps of size 16x16 from the 128x128 surface, which are concatenated into a 3-dimensional latent representation of shape 16x16x32, effectively increasing the width dimension. The latent representation can be partitioned into two blocks of size 16x16x16, representing the mean and log-variance of the learned features. Assuming priors of diagonal Gaussian distributions, we can take samples from the standard Gaussian distribution and use the reparameterisation trick to optimise for the variational learning.

The Multi-kernel Convolution Block is the elementary building block in the first section of the VAE network. The block is structured to first perform normalisation and activation first, then conv1D operations are performed on a list of kernel sizes, where the results are then concatenated. An example Multi-kernel Convolution Block with the list of kernel sizes [1, 3, 5] is illustrated in figure 9.

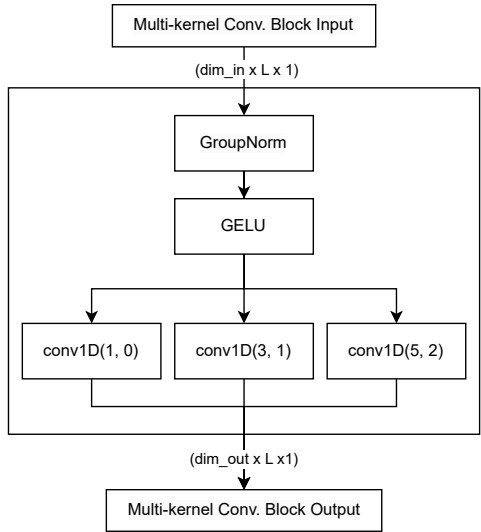

Figure 9: **An Example of Multi-kernel Convolution Block**

# E    UNet

The UNet model used consists of four down and four up blocks. Each block consists of eight sequential ResNet blocks, followed by a single upsampling or downsampling block to change the number of channels. The third down block and the second up block are enriched with cross-attention layers, one applied after each ResNet block.

Each ResNet block transforms an input tensor through operational layers including normalisation, swish non-linearity, upsampling, two convolutional layers and time-embedding projection, culminating in an output tensor formed by summing the input tensor and the output of the second convolutional layer.

Attention layers, integral to the refinement of the feature representation, are incorporated in the third down and second up blocks, each consisting of eight 64-dimensional attention heads, executed by a scaled dot product attention mechanism without dropout and bias in the projection layers.

The UNet input and output contain 16 channels, each containing 16 x 16 arrays. The down blocks have channel dimensions of [256,256,512,512] and the up blocks mirror the encoder blocks in reverse order. In the forward process, we use N = 1000 steps.

# F    Extra Results

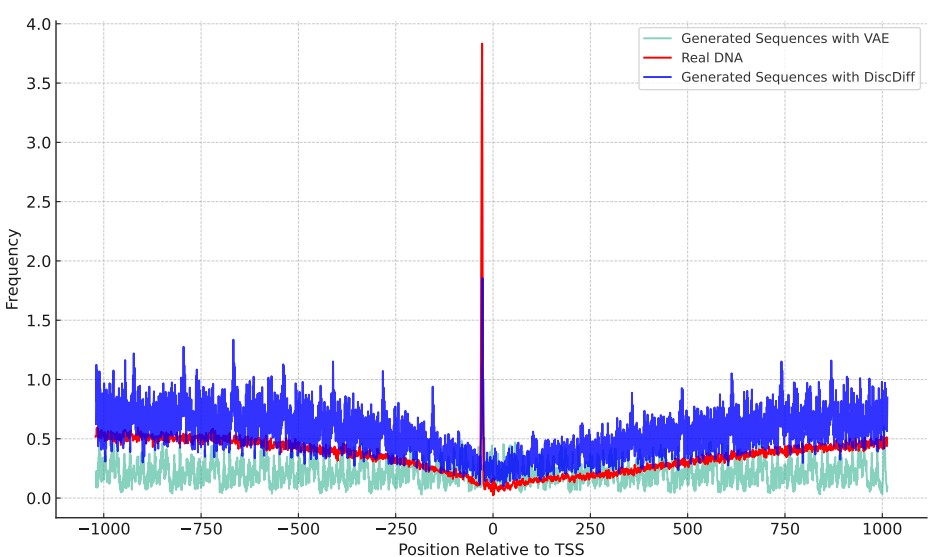

Figure 10: **Compare the TATA-Box Distribution of VAE, DiscDiff, and Real DNA Sequences**

