# OpenReview forum: "Latent Diffusion Model for DNA Sequence Generation"
_NeurIPS.cc/2023/Workshop/AI4Science — NeurIPS2023-AI4Science Poster_

### Official Review · Reviewer_UFQG · 2023-10-06
**A Novel Approach for DNA Sequence Synthesis**

**Rating:** 6
**Confidence:** 4

**Review:**

**Strength**
1. The paper is well-written with a good motivation that formulates the proposed framework.
2. The downstream research for the latent diffusion model is an attractive problem worthy of exploration. The paper shows promising evaluations of the diffusion synthesis of DNA sequences.
3. The involved metrics like FReD and Motif Distribution are creative.

**Weakness**
1. It will contribute to the work if the authors elaborate more on the superiority of the framework compared to autoregressive-based generative models.
2. The paper should consider utilizing pre-trained knowledge in open-sourced large generative models (e.g., StableDiffusion, GPT2), which would further boost the performance of this framework.

---

### Official Review · Reviewer_x42L · 2023-10-21
**Promising with potential for many applications**

**Rating:** 7
**Confidence:** 4

**Review:**

The authors present DiscDiff, which combines a VAE with a diffusion model to generate synthetic DNA sequences. They also develop methods to evaluate the quality of the generated sequences. Overall, the work is promising. Below are a few comments and suggestions:

- for Sei, it would be a good idea to project the Sei embeddings of real and synthetic sequences into a UMAP for visualization
- "H3Kxxme3 markers are closely linked to promoter activity" - this is not true, while H3K4me3 is linked to promoter activity, H3K27me3 is linked to repressed chromatin, H3K27ac is linked to enhancer/promoter activity, and H3K9me3 is linked with heterochromatin which is repressed
- It would be useful to characterize the sequence composition (e.g. GC content, k-mer composition) of real vs synthetic sequences
- For the promoter-gene sequences, evaluation can be stratified into promoter and gene body separately
 - designing enhancer sequences would be a good application for the method (see https://www.biorxiv.org/content/10.1101/2022.07.26.501466v1)

---

### Meta-Review · Area_Chair_G2fG · 2023-10-26

**Recommendation:** Accept (Oral)
**Confidence:** 3

**Metareview:**

The paper introduces "DiscDiff," a novel latent diffusion model specifically designed for generating synthetic DNA sequences. This model attempts to harness the power of continuous diffusion models, previously used primarily in image generation, for discrete data like DNA sequences by embedding these sequences into a continuous latent space using an autoencoder. Alongside the model, a new metric, Fréchet Reconstruction Distance (FReD), is proposed to assess the quality of the generated DNA sequences. The work further contributes a significant dataset of promoter-gene sequences from various species.

Both reviewers acknowledge the potential and novelty of the proposed approach. The key strengths highlighted include:

- Well-structured and motivated paper.
- Introduction of creative metrics such as FReD.
- The generative approach's promise as shown in evaluations.
- Significant contribution in the form of a cross-species dataset.

There are also suggestions from reviewers:

- A need for a visualization of Sei embeddings for better interpretability.
- Corrections and clarifications on chromatin markers.
- A suggestion to analyze the sequence composition of real vs. synthetic sequences.
- A stratified evaluation for promoter and gene body sequences.
- Recommendations to extend applications, such as designing enhancer sequences.
- A deeper discussion comparing DiscDiff to autoregressive-based generative models.
- Exploration of leveraging pre-trained models to further enhance DiscDiff's performance.

The paper presents an innovative approach to a complex problem and has the potential to open up new avenues in the field of synthetic DNA sequence generation. While there are valid concerns and suggestions from the reviewers, the overall promise and contributions of the work, especially when combined with potential revisions, make it a valuable addition to the literature. The authors should be encouraged to take the feedback into account for the final version of the paper.